

# LncRNACNVIntegrateR: a novel framework for correlating long non-coding RNAs with copy number variation abnormalities and disease progression

Neetu Tyagi[1,2], Shikha Roy[2] and Dinesh Gupta[2]

[1] Regional Centre for Biotechnology, Faridabad, Haryana, India
[2] Structural and Computational Biology Group, International Centre for Genetic Engineering and Biotechnology (ICGEB), New Delhi, Delhi, India

Corresponding author
Dinesh Gupta, dinesh@icgeb.res.in

## ABSTRACT

Understanding complex biological systems and disease mechanisms necessitates the integration of multiple molecular layers, making multi-omics data integration a cornerstone of modern biomedical research. By combining datasets from different omics domains, researchers can uncover intricate molecular relationships, discover robust biomarkers, and advance precision medicine. Despite advancements in high-throughput technologies that have increased the availability of multi-omics datasets, challenges such as sample consistency and the development of reliable analytical frameworks hinder their full potential. Addressing these challenges is crucial for achieving a comprehensive understanding of biological systems and leveraging multi-omics data to drive breakthroughs in healthcare. lncRNACNVIntegrateR is an R package that facilitates multi-omics data integration to explore the interplay between long non-coding RNAs (lncRNAs) and copy number variations (CNVs). The package integrates transcriptomic data, CNV profiles, and clinical information from matched samples, providing a complete pipeline for data preprocessing, lncRNA-CNV correlation analysis, and identification of CNV-driven prognostic signatures. Additionally, the package supports the construction of risk score models based on CNV-associated lncRNAs and functional enrichment analyses to reveal the role of corresponding target genes in disease progression. We validated lncRNACNVIntegrateR using The Cancer Genome Atlas (TCGA) Glioblastoma (GBM) and Colorectal Adenocarcinoma (COAD) datasets. The risk score models developed by the package demonstrated promising predictive performance, achieving an area under the receiver operating characteristic curve (AUC) of 0.80 for GBM and 0.71 for COAD. Functional enrichment analyses further highlighted the biological significance of the identified prognostic CNV-driven lncRNA signatures, providing insights into disease progression, risk stratification, and potential therapeutic targets to support clinical decision-making and personalized treatment approaches.

## INTRODUCTION

Advancements in high-throughput sequencing have revolutionized biomedical research, enabling multi-level integration of genomic, transcriptomic, methylomic, and epigenomic data to unravel complex diseases and multidimensional biological patterns (*Lee, 2023*). To comprehensively understand complex biological processes, it is crucial to adopt an integrative approach combining multi-omics data and exploring the interrelationships among biomolecules and their functions. With the growing availability of multi-omics data from numerous samples, several tools and methods have been developed to integrate and interpret data for various applications, including disease subtyping, biomarker prediction, and data analysis (*Subramanian et al., 2020*). Numerous studies have shown that long non-coding RNAs (lncRNAs) are crucial regulatory molecules that can be targeted to modulate cellular physiology and functions in the pathophysiological progression of tumors, as well as in other diseases such as autoimmune diseases and cardiovascular diseases (*Hajjari, Khoshnevisan & Shin, 2014*; *Kumar & Goyal, 2017*; *Lorenzen & Thum, 2016*; *Ng et al., 2013*; *Ounzain et al., 2015*; *Wu et al., 2015*). Integrative analyses of various omics datasets have highlighted the deregulation of lncRNAs due to copy number variations (CNVs), drawing significant attention to their roles.

lncRNAs are >200-nucleotide RNA molecules that lack protein-coding potential (*Beermann et al., 2016*). They are key regulators in human diseases, particularly cancer, where they influence critical processes such as cell proliferation, signaling pathways, angiogenesis, and metastasis, driving various stages of cancer progression (*Kondo, Shinjo & Katsushima, 2017*; *Ling, Fabbri & Calin, 2013*). Additionally, lncRNAs can function as decoys or sponges, sequestering regulatory factors and miRNAs to modulate gene expression (*Morriss & Cooper, 2017*). LncRNAs regulate gene expression by modifying chromatin, influencing allelic imprinting, and controlling post-transcriptional and post-translational processes (*Mercer, Dinger & Mattick, 2009*; *Wilusz, Sunwoo & Spector, 2009*).

CNVs are genetic alteration involving duplications or deletions of DNA segments ranging from 1 kb to 3 Mb (*Nakamura, 2009*). In complex diseases like cancer, CNVs can activate oncogenes or delete tumor suppressor genes, affecting cellular adhesion, communication, and function (*Conrad et al., 2010*). CNVs contribute to individual variability and were initially considered protective by creating redundancy. However, many CNVs are now recognized as harmful, being associated with various diseases, including neurological and developmental disorders (*Morrow, 2010*; *Valsesia et al., 2013*).

By altering the expression of genes, including lncRNAs, CNVs can disrupt normal cellular functions and contribute to disease progression. The interaction between CNVs and lncRNAs is crucial not only in cancer (*Ning et al., 2021*; *Tian & Luo, 2022*; *Wang et al., 2020*; *Zheng et al., 2020*; *Zhong et al., 2021*) but also in various other diseases (*Lu et al., 2023a, 2023b*). This interplay can disrupt key genes and pathways contributing to disease development and progression and underscoring the broader impact of CNVs and lncRNAs association on human health.

The importance of novel tools for multi-omics data integration lies in their ability to address the complexities of harmonizing diverse datasets, enhancing our understanding of biological systems and improving clinical applications. For instance, the mixOmics package (*Rohart et al., 2017*) enables the multivariate analysis and integration of high-throughput 'omics data, facilitating the discovery of molecular relationships. TimeOmics (*Bodein et al., 2022*) offers a comprehensive framework for analyzing longitudinal multi-omics data, crucial for studying dynamic biological processes. Holomics (*Munk et al., 2024*) provides an accessible workflow for single-omics analysis, addressing challenges in data scale and complexity. MOVICS (*Lu et al., 2021*) focuses on cancer subtyping through multi-omics integration, advancing personalized medicine. These tools exemplify the critical advancements needed to streamline data integration and interpretation, ultimately leading to better disease understanding and therapeutic strategies.

To address the growing demand for user-friendly tools in multi-omics data integration, we developed lncRNACNVIntegrateR, an R package, to explore the relationship between lncRNAs and CNV abnormalities for disease diagnosis and prognosis. We tested lncRNACNVIntegrateR on The Cancer Genome Atlas (TCGA) datasets for Glioblastoma (GBM) and Colorectal Adenocarcinoma (COAD), identifying several key CNV-driven lncRNA prognostic biomarkers. Furthermore, we validated the risk score model, and functional enrichment analysis revealed the roles of these biomarkers in disease progression, providing insights into their prognostic potential and supporting clinical decision-making for personalized treatment.

## METHODS

### Design and implementation

The structure and design of the lncRNACNVIntegrateR package is modular and user-centric, ensuring ease of use and flexibility for integrating and analyzing multi-omics data. The package is organized into several core functions, each performing specific tasks such as data input, preprocessing, correlation analysis, risk score model development, and functional enrichment analysis. This modular approach allows users to utilize each function independently or as a comprehensive workflow. Figure 1 illustrates the end-to-end workflow implemented in lncRNACNVIntegrateR, highlighting the major data integration and analysis steps. The analysis begins with three essential input files consisting of the following datasets:

(1) **Expression profiles:** Provided as a raw count matrix or a tab-delimited file. Rows represent genes (including lncRNAs), and columns represent samples. The first row should include sample identifiers (header), followed by gene identifiers and their corresponding expression values.

(2) **Absolute CNV profiles:** A matrix or tab-delimited file where rows correspond to CNV features and columns to samples. The header includes sample IDs, followed by rows
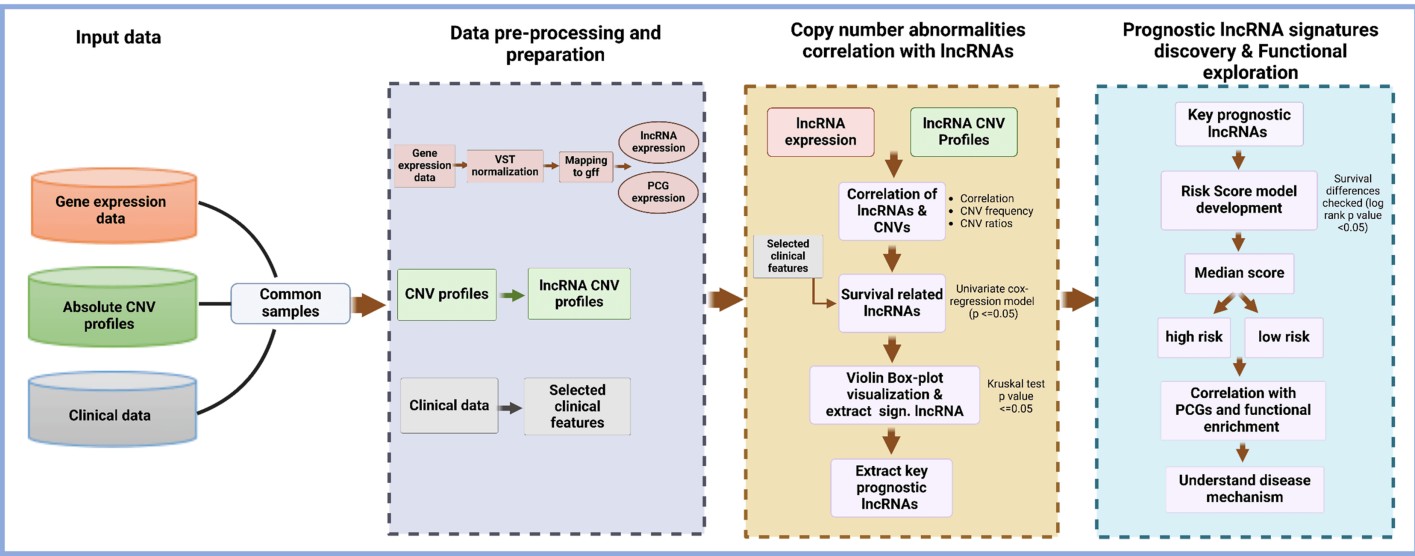

**Figure 1 Overview of the lncRNACNVIntegrateR workflow.** A schematic representation of the analytical workflow, starting from raw multi-omics input and proceeding through integration, biomarker discovery, risk modeling, and functional interpretation.

with CNV identifiers and their absolute copy number values (indicating amplifications or deletions).

(3) **Clinical data:** A tab-delimited file with clinical annotations. Rows represent samples and columns represent clinical variables, including at minimum: Sample_ID, Survival_Time, and Status.

These datasets are processed through sequential stages of data harmonization, correlation analysis, survival modelling, and downstream functional exploration.

### Core functionalities of the package

The lncRNACNVIntegrateR package provides a comprehensive suite of functions, each corresponding to a key step in the analysis pipeline (Fig. 1). These include:

**Data preprocessing:** The function automatically harmonizes sample IDs across the expression, CNV, and clinical datasets by identifying the common samples shared among all three. It then proceeds with downstream analysis using only these matched samples, ensuring consistency and compatibility across data types.

**Correlation analysis:** Computes statistical associations between lncRNA expression and CNV status to identify CNV-driven lncRNAs.

**Survival association testing:** Evaluates the prognostic significance of candidate lncRNAs using clinical survival data.

**Signature identification:** Filters for the most informative CNV-deregulated lncRNAs based on statistical significance and biological relevance.

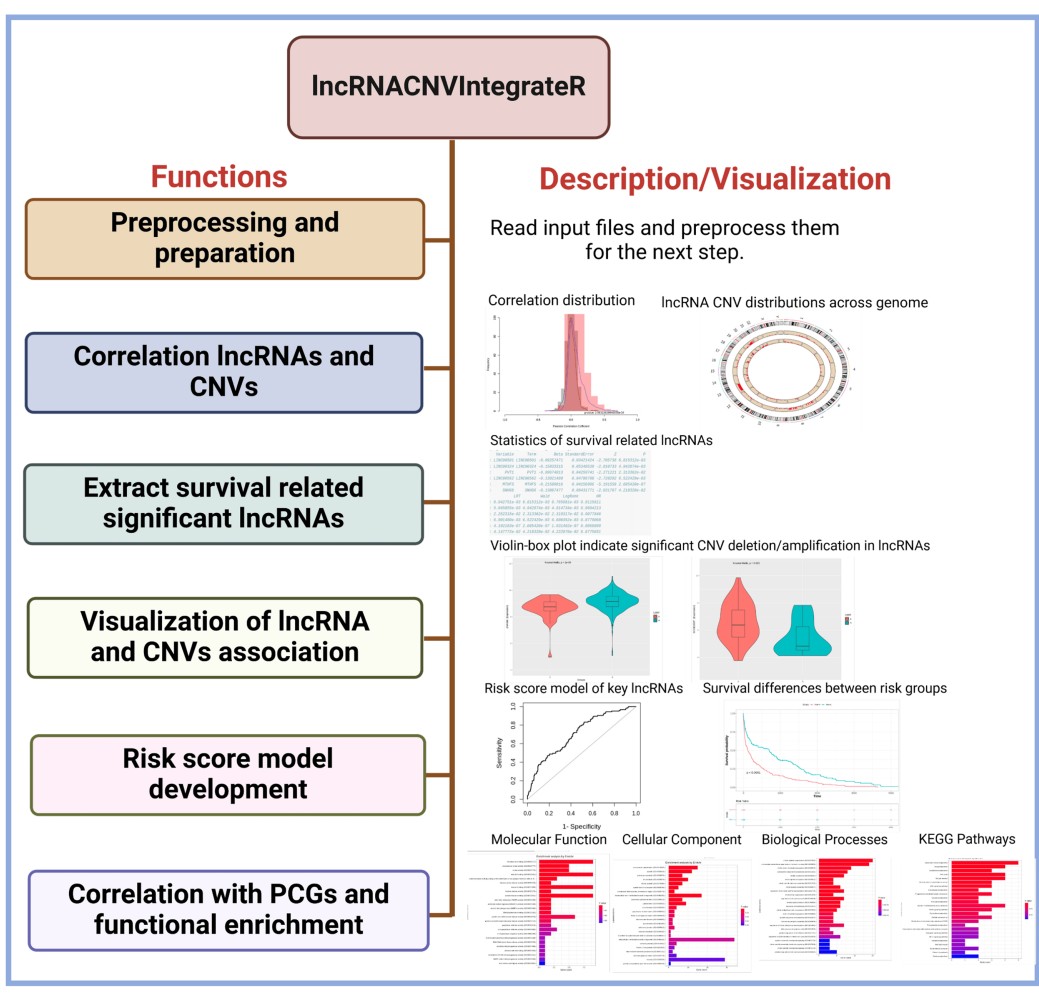

**Figure 2 Core functions in the lncRNACNVIntegrateR package.** A function-level diagram showing the main analytical modules provided in the package, corresponding to each major step in the workflow.

**Risk score modelling:** Constructs and validates a prognostic model using selected lncRNA signatures.

**Functional enrichment analysis:** Investigates the downstream mRNA targets of identified lncRNAs to interpret their indirect roles in biological pathways and processes.

Each of these functionalities is implemented as a standalone function, allowing users to use each function in a standalone mode in their data analysis (as shown in Fig. 2). The structure and design of the lncRNACNVIntegrateR package is modular and user-centric, ensuring ease of use and flexibility for integrating and analyzing multi-omics data.

The lncRNACNVIntegrateR package processes the three primary input files: expression profiles, absolute CNV profiles, and clinical data (as discussed above) through various dedicated functions. It facilitates comprehensive analysis and integration of the data, encompassing several key tasks. These tasks include data preprocessing and preparation, evaluating the correlation between lncRNA expression and CNVs, and identifying

survival-associated lncRNAs. The package further identifies significant CNV-deregulated lncRNA signatures, constructs a risk score model for these signature lncRNAs, and performs functional enrichment analysis on the target genes of the identified lncRNA signatures. Detailed descriptions of the functions provided by this package are outlined below.

## Input functions

To streamline data handling, the package provides dedicated input functions for each data type:

- input_expr for loading raw gene expression count matrices (whole transcriptome data),
- input_cnv for importing absolute CNV profiles, and
- input_clin for clinical data, including survival time and event status.

In addition, the package supports annotation using the GTF file *via* the input_gtf function (sourced from GENCODE version 22) and retrieves a curated list of lncRNAs using the input_lncRNAs function. The 'input_lncRNA_positions' function helps incorporate genomic positions of lncRNAs for downstream analyses and is available in the example extdata directory of the package.

These datasets serve as inputs for the analysis and are passed into the package using the corresponding input functions. For the GBM and COAD case study, gene expression and clinical data were obtained from TCGA *via* the GDC portal (https://gdc.cancer.gov/), while absolute CNV data were downloaded from the UCSC Xena Browser (http://xena.ucsc.edu/). These datasets serve as inputs for the analysis and are passed into the package using the corresponding input functions.

## Pre-processing

The 'preprocessing_and_preparation' function conducts preprocessing on individual data types. The first step involves identifying common samples across the three data types, extracting the relevant information from each data type for these common samples, and then proceeding with the preprocessing. For gene expression data, first, data normalization is performed using the DESeq2 Variance Stabilizing Transformation (VST) (*Love, Huber & Anders, 2014*). Further, the lncRNAs and protein-coding genes (PCGs) are mapped using GTF information (obtained from gencode version 22, https://www.encodeproject.org/files/gencode.v22.annotation). Genes categorized in the GTF file under labels such as "processed_transcripts," "3prime_overlapping_ncRNA," "sense_intronic," "sense_overlapping," "antisense," and "lincRNA" are recognized as lncRNAs, whereas those labeled as "protein_coding" are identified as PCGs. Following the extraction of lncRNAs and PCGs expression profiles, the CNV profiles for the lncRNAs are then extracted from the total absolute CNV data. Important parameters, including survival-related information, are extracted from the clinical data.

Expression data undergoes lncRNA and PCG annotation based on the GTF file. Genes annotated as "processed_transcript," "3prime_overlapping_ncRNA," "sense_intronic," "sense_overlapping," "antisense," and "lincRNA" are considered lncRNAs, while those

labeled "protein_coding" are designated as PCGs. Expression data is normalized using the VST method from the DESeq2 package (*Love, Huber & Anders, 2014*). Simultaneously, lncRNA-specific CNV profiles are extracted from the absolute CNV dataset. Clinical parameters such as survival time and status are also extracted at this stage.

### lncRNA-CNV correlation

After the data preprocessing and preparation step, the processed lncRNA expression and CNV profiles are utilized to compute the Pearson correlation coefficient (PCC) for each lncRNA using the 'Correlation_lncRNAs_and_CNVs' function. The correlation distribution is then plotted against a random distribution to evaluate any correlation. Subsequently, prognosis-related lncRNA signatures are extracted using a univariate Cox proportional risk regression model with a *p*-value cut-off <0.05 under function 'extract_survival_related_significant_lncRNA'. Significant lncRNAs exhibiting amplification or deletion are further identified using a violin-box plot based on the Kruskal-Wallis test (cut-off *p*-value < 0.05).

### Risk score model construction

The list of screened lncRNAs displaying significant CNV amplification/deletion, identified through the Kruskal-Wallis test in the 'visualization_of_lncRNAs_and_CNVs_association' function, is refined using a *p*-value cut-off <0.05. Key prognostic lncRNAs (pr-lncRNAs) are selected to construct a risk score model using the PredictABEL (a R package available at https://cran.r-project.org/package=PredictABEL, version 1.2-4). With the identified key lncRNAs, the model can stratify patients into two risk groups using their respective risk scores. This functionality is implemented through the 'Risk_score_model_development' function. A survival plot is created to determine if these two groups differ in survival probabilities (*p*-value < 0.05).

The list of screened lncRNAs showing significant CNV amplification/deletion, determined by the Kruskal-Wallis test, was further refined using a multivariate Cox regression model with a *p*-value < 0.05. Key prognostic lncRNAs (pr-lncRNAs) were chosen to build a risk score model using the PredictABEL R package (https://cran.r-project.org/package=PredictABEL). Based on these key lncRNAs, this model can classify patients into two risk categories using their respective risk scores. A survival plot was created to determine if these two groups differ in survival probabilities (*p*-value < 0.05).

### Functional enrichment analysis

To explore the function of the identified pr-lncRNAs, the tool extracts highly correlated genes associated with these pr-lncRNAs. To analyze the functional enrichment of PCGs correlated with these lncRNAs, we utilize the enrichR package (*Kuleshov et al., 2016*) within the 'Correlation_with_PCGs_and_functional_enrichment' function.

## RESULTS

To demonstrate the utility of lncRNACNVIntegrateR, we performed an analysis of GBM and COAD cancer datasets. The gene expression and clinical data sets were obtained from TCGA, and absolute CNV calls were sourced from UCSC Xena, respectively.

## Case study 1: identification of CNV-driven lncRNA signatures in glioblastoma using lncRNACNVIntegrateR

To demonstrate the utility of lncRNACNVIntegrateR, we analyzed multi-omics data from GBM, the most aggressive primary brain tumor with a 5-year survival rate of 5.8% (*Zhu et al., 2021*). The lncRNACNVIntegrateR pipeline was applied to integrate gene expression, CNV, and clinical data from the GBM dataset, accessed *via* the 'input_expr', 'input_cnv', and 'input_clin' functions. Data preprocessing was performed using the 'preprocessing_and_preparation' function, which extracted lncRNA and PCG expression profiles mapped to a GTF reference file.

The 'Correlation_lncRNAs_and_CNVs' function was used to evaluate associations between lncRNA expression and CNV profiles. This analysis revealed statistically significant correlations ($p < 0.05$), which were visualized as correlation distribution plots. The Pearson correlation distribution is shown in Fig. 3A, and the genome-wide distribution of lncRNA CNVs is presented in Fig. 3B. Next, the 'extract_survival_related_significant_lncRNA' function was applied to identify lncRNAs significantly associated with patient survival. The 'visualization_of_lncRNA_and_CNVs_association' function further confirmed CNV alterations, amplifications and deletions for these lncRNAs, as illustrated by representative violin plots.

By applying the Kruskal-Wallis test to compare CNV groups (diploid *vs.* amplified/deleted), and selecting those with statistically significant differences, we identified 21 lncRNAs significantly associated with both CNV alterations and patient survival. These include C1orf229, C3orf35, C9orf147, FENDRR, FTX, HCG18, LINC00115, LINC00265, LINC00271, LINC00324, LINC00354, LINC00641, LINC00645, LINC00667, LINC00676, LINC00858, LINC00907, LINC00963, LINC00970, MIAT, NEAT1, PCA3, RNU6ATAC35P, SNHG6, SNHG7, and SNX29P2.

A risk score model was then developed using the 'Risk_score_model_development' function. The model achieved a strong predictive performance with an AUC of 0.794 (95% CI [0.722–0.867]), as shown in Fig. 3C. Further, Kaplan-Meier survival analysis based on the expression of these 21 lncRNA signatures stratified GBM patients into high-risk and low-risk groups, demonstrating a statistically significant difference in overall survival ($p < 0.05$, Fig. 3D).

Using the full functionality of lncRNACNVIntegrateR, integration analysis of GBM datasets identified 21 lncRNA signatures for GBM as described above. Among these, LINC00115, LINC00324, LINC00641, LINC00963, MIAT, NEAT1, SNHG6, and SNHG7 play significant roles in glioma progression and therapy, suggesting their potential as biomarkers and therapeutic targets for glioblastoma (*Chen, Lin & Li, 2023*; *Chen et al., 2020*; *Liang et al., 2022*; *Nie et al., 2021*; *Rao et al., 2024*; *Tang et al., 2019*; *Yang et al., 2020*; *Ye et al., 2020*). Thereby validating the findings and demonstrating the tool's reliability in this study. In addition, several other lncRNAs identified in this study, including LINC00271, LINC00265, LINC00858, and LINC00667, have also been implicated in the pathogenesis of various cancer types beyond GBM. The risk score model achieved an exceptional AUC of 0.794 with a 95% Confidence Interval [0.722–0.867]. The survival

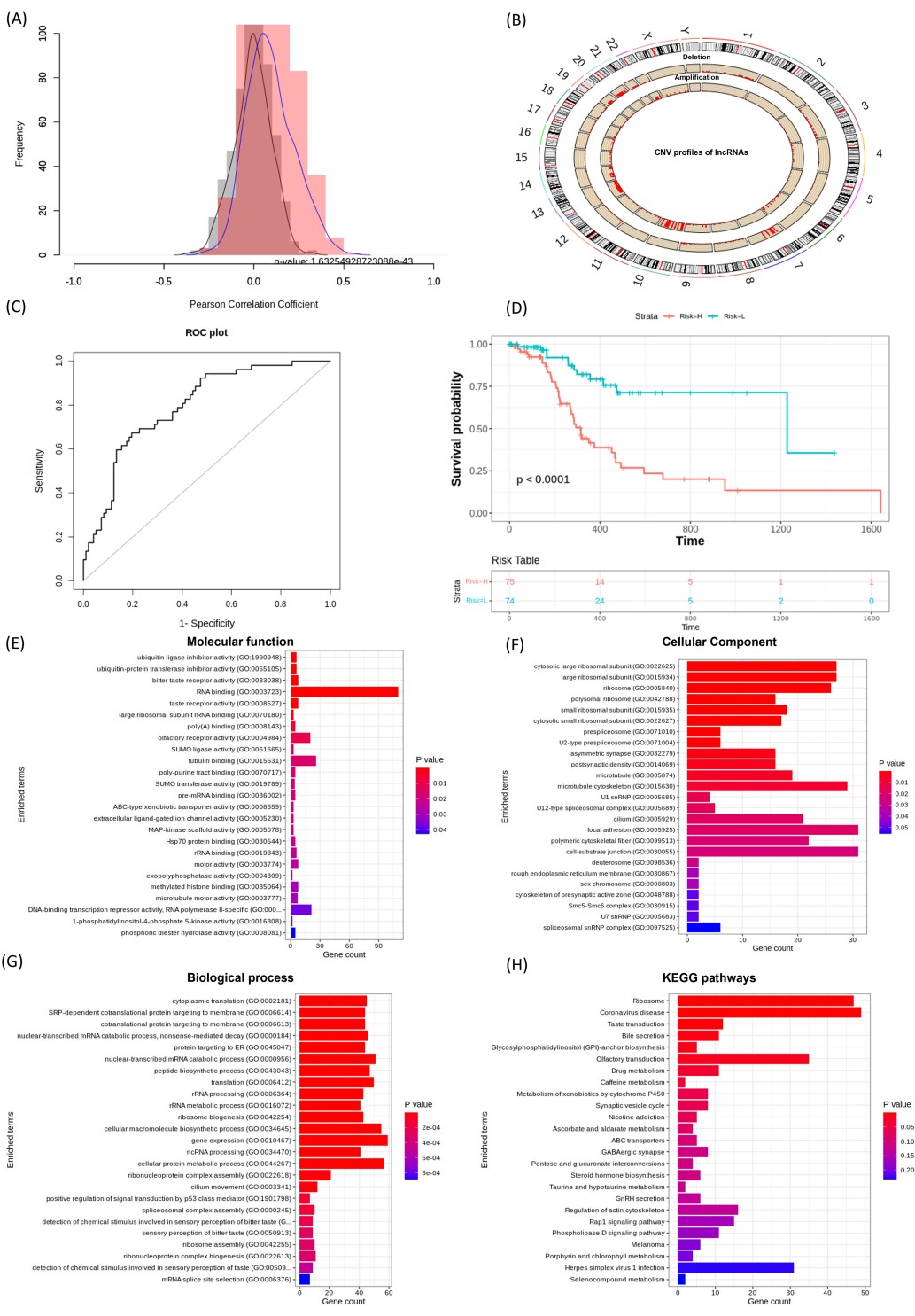

**Figure 3 Performance of lncRNACNVIntegrateR on the GBM dataset.** (A)–(H) illustrate the analysis workflow: (A) Correlation between lncRNAs and CNVs, (B) CNV distribution across the genome, (C) risk score plot for the final lncRNA signatures influenced by significant CNV changes, (D) survival analysis comparing high-risk and low-risk patient groups based on the risk score model, and (E–H) functional enrichment analysis linking lncRNAs to disease mechanisms.

analysis also effectively distinguished high-risk and low-risk groups and verified the model's accuracy.

Functional enrichment analysis (using function 'Correlation_with_PCGs_and_functional_enrichment') of the top correlated PCGs with the identified lncRNAs revealed significant enrichment in Gene Ontology (GO) categories, providing valuable insights into their potential biological functions. Among these, several key Biological Processes (BP) were identified to be enriched, which include ncRNA processing, p53-mediated signal transduction, nonsense-mediated mRNA decay, and regulation of gene expression, as illustrated in Fig. 3G. Corresponding Cellular Component (CC) and Molecular Function (MF) categories were also enriched, supporting the regulatory influence of these lncRNAs on diverse cellular mechanisms. Furthermore, Kyoto Encyclopedia of Genes and Genomes (KEGG) pathway analysis demonstrated that these lncRNA-targeted genes are involved in critical signaling pathways, such as the Rap1 signaling pathway, regulation of the actin cytoskeleton, and ABC transporter pathways (Fig. 3H). Collectively, these results highlight the involvement of lncRNA-associated PCGs in essential oncogenic processes, reinforcing their potential as biomarkers and therapeutic targets in GBM.

## Case study 2: identification of CNV-driven lncRNA signatures in colon adenocarcinoma using lncRNACNVIntegrateR

To further evaluate and demonstrate the utility of lncRNACNVIntegrateR, we analyzed multi-omics data for COAD dataset, the predominant subtype of colorectal cancer, which ranks as the fourth most diagnosed cancer and second leading cause of cancer-related deaths in the United States (Benson et al., 2018). Cancer prevention, screening, and treatment advances have significantly reduced CRC incidence and mortality rates (Edwards et al., 2010). However, the prognosis for patients with advanced colon cancer remains poor (Sadanandam et al., 2013), with 90% of these cases being COAD (Hajmanoochehri et al., 2014).

Effective prognostic stratification is essential for improving patient outcomes in COAD. Using lncRNACNVIntegrateR, we integrated gene expression, CNV, and clinical data from COAD dataset. Significant correlations between lncRNA expression and CNV profiles were observed ($p < 0.05$), as shown in the correlation distribution (Fig. 4A) and the genome-wide CNV distribution plot (Fig. 4B). Applying this framework, the analysis identified 14 distinct CNV-driven lncRNA signatures, including SNHG7, TMEM191A, PCAT6, MIR210HG, MIAT, LINC00847, LINC00486, LINC00115, GAS5, FENDRR, FAM85B, C7orf66, C10orf95-AS1, and BCYRN1. A risk score model achieved an AUC of 0.707 (95% CI [0.631–0.782]), demonstrating reliable prognostic performance (Fig. 4C). Further, based on these 14 lncRNA signatures, the stratification of COAD patients into high- and low-risk groups (Fig. 4D), with Kaplan-Meier survival analysis confirming significant survival differences.

Eight lncRNAs, including SNHG7, PCAT6, MIR210HG, MIAT, LINC00115, GAS5, FENDRR, and BCYRN1 have been previously linked to colorectal cancer progression,

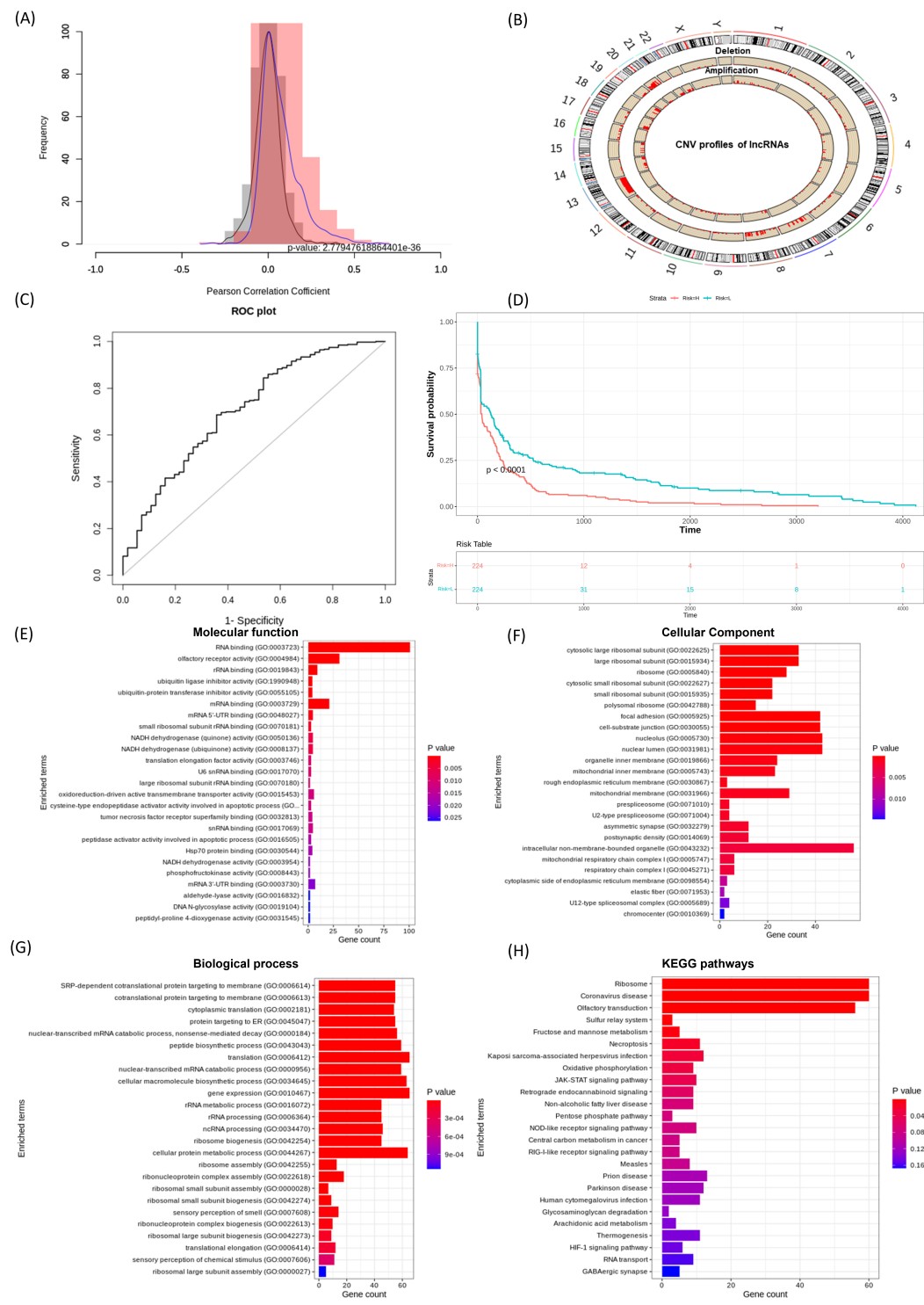

**Figure 4 lncRNACNVIntegrateR performance on the COAD dataset.** (A–F) The analysis workflow: (A) lncRNA-CNV correlation, (B) CNV distribution across the genome, (C) risk score plot for the final lncRNA signatures driven by significant CNV changes, (D) survival analysis comparing high-risk and low-risk patient groups based on the risk score model, and (E–H) functional enrichment analysis to associate lncRNAs with disease mechanisms.

prognostic implications, and therapeutic response (*Pan et al., 2023*; *Han et al., 2024*; *Ruan et al., 2019*; *Liu et al., 2018*). *Feng et al. (2020)*, *Xie et al. (2022a)*, *Yang et al. (2023)*, *Gu et al. (2018)* corroborating the tool's ability to identify clinically relevant biomarkers. These results underscore lncRNACNVIntegrateR's efficacy in uncovering CNV-driven lncRNA signatures with prognostic and therapeutic potential in COAD.

Figures 4E–4H showcases the results of functional enrichment analysis for the top correlated genes associated with these signature lncRNAs, emphasizing their roles in COAD tumorigenesis. Functional enrichment analysis revealed significant involvement of the correlated genes in key biological processes such as ncRNA processing, nonsense-mediated mRNA decay, translation, and gene expression. KEGG pathway analysis further highlighted enrichment in critical cancer-related pathways, including JAK-STAT signaling, necroptosis, viral infection, and central carbon metabolism in cancer. These findings demonstrate the utility of lncRNACNVIntegrateR in identifying prognostic lncRNA signatures and uncovering their functional roles in colorectal adenocarcinoma, supporting their relevance in disease progression and potential therapeutic targeting.

## DISCUSSION

The lncRNACNVIntegrateR package offers a robust, user-friendly framework for integrating CNV and lncRNA expression data to identify prognostic biomarkers. We demonstrated its performance on two common cancers like GBM and COAD. By automating multi-omics data analysis, the tool streamlines the identification of CNV-driven lncRNA signatures, risk score modeling, and functional enrichment, making it accessible to researchers with varying computational expertise. Its ability to uncover both established and novel lncRNAs underscores its potential to advance biomarker discovery.

In this study, we leveraged the full capabilities of the lncRNACNVIntegrateR package to systematically analyze the interplay between CNVs and lncRNA expression in GBM and COAD datasets. GBM is one of the most aggressive and treatment-resistant forms of brain cancer; similarly, CAOD is also the most common form of Colorectal Cancer. Through lncRNACNVIntegrater analysis, we identified 21 lncRNAs exhibiting CNV-associated expression alterations in GBM and 14 in COAD.

Among these, several lncRNAs have been previously implicated in both the diseases. Their recurrence in our study provides biological validation of our approach and highlights the robustness of lncRNACNVIntegrateR in discovering biologically relevant signatures. The remaining lncRNAs, which have not yet been characterized in GBM and COAD, may serve as novel candidates for future functional validation.

Compared to existing methods, lncRNACNVIntegrateR demonstrates superior accuracy and functionality by systematically identifying CNV-driven lncRNAs and constructing risk models integrating genomic and transcriptomic layers. *Lei et al. (2025)* successfully built a prognostic model using cuproptosis-associated lncRNAs based on co-expression with known cuproptosis genes; however, its focus remained limited to a specific cell death pathway. Moreover, the associations were inferred solely from transcriptomic data, without investigating potential genomic drivers such as CNVs that

may underlie lncRNA dysregulation. In another study, although the artificial intelligence prognostic signature (AIPS) model demonstrated strong prognostic performance and integrated gene-level CNV and methylation data, it primarily focused on protein-coding genes (*Jiang et al., 2024*). It did not systematically investigate CNV-driven regulatory mechanisms involving lncRNAs or construct a structured multi-omics framework centered on non-coding RNA regulation. Another study constructed a prognostic model for GBM using four m6A-related lncRNAs (*AC005229.3, SOX21-AS1, AL133523.1*, and *AC004847.1*) identified *via* LASSO regression (*Xie et al., 2022b*).

In context of COAD, *Lin, Liu & Xu (2025)* identified CNV-driven lncRNAs and constructed a ceRNA network elucidating their involvement in essential biological processes. LINC00941 amplification was notably linked to poor prognosis, and potential drug targets were proposed. However, no prognostic risk score model was developed to evaluate the clinical utility of these CNV-lncRNAs (*Lin, Liu & Xu, 2025*). Another study established a prognostic model based on 19 HRR-related lncRNAs using TCGA-COAD data, which effectively stratified patients by survival risk and demonstrated strong predictive power. Functional enrichment further linked the model to the mitogen-activated protein kinase (MAPK) signaling pathway and homologous recombination deficiency (HRD), emphasizing its potential for precision medicine in colon cancer (*Tang et al., 2022*).

While numerous studies have developed prognostic models in GBM and COAD, mostly focused on specific transcriptomic features such as cuproptosis, amino acid metabolism, m6A modification, or homologous recombination repair without examining the genomic mechanisms regulating lncRNAs. Notably, none have systematically investigated CNV-mediated regulation of lncRNAs or provided dedicated tools for such integrative analysis. Addressing this gap, we developed lncRNACNVIntegrateR, which identifies CNV-driven lncRNAs and leverages them for robust prognostic modelling. By enabling user-friendly multi-omics integration and downstream analysis, the tool advances biomarker discovery and supports broader applications in precision oncology.

This study has certain limitations. The evaluation of lncRNACNVIntegrateR was limited to publicly available tumor datasets from GBM and COAD, restricting its current assessment to cancer-specific contexts. Although the tool is designed for broader use, its performance on non-tumor datasets has not yet been examined. Moreover, while the current framework focuses on identifying CNV-driven lncRNAs and associated prognostic signatures, incorporating advanced CNV visualization tools and additional regulatory layers could further improve its interpretability and expand its applications.

## CONCLUSION

Multi-omics dataset integration tools are crucial for comprehensively studying complex biological systems and identifying potential biomarkers. To meet the demand for user-friendly tools, we developed an R package, lncRNACNVIntegrateR, that integrates transcriptome and CNV data, enabling researchers to identify significant CNV-driven lncRNAs. This package facilitates exploring and developing personalized treatment strategies and clinical decision-making by providing deeper insights into the interaction

between lncRNAs and CNVs. We validated lncRNACNVIntegrateR by analyzing two TCGA cancer datasets, which enabled the shortlisting of promising biomarkers, demonstrated effective risk stratification, and provided robust survival analysis. The lncRNACNVIntegrateR can identify several reported as well as novel biomarkers in the datasets, and hence, the identified potential biomarkers may also be validated experimentally. Based on its robust validations, lncRNACNVIntegrateR serves as a valuable tool for integrating transcriptome and CNV datasets, enabling the discovery of novel and meaningful correlations between CNVs and long non-coding RNAs (lncRNAs) with significant prognostic potential.

### Funding

This work was supported by the bioinformatics infrastructure grant from the Department of Biotechnology, Government of India (no. BT/PR40151/BTIS/137/5/2021). The funders had no role in study design, data collection and analysis, decision to publish, or preparation of the manuscript.

### Grant Disclosures

The following grant information was disclosed by the authors:
Department of Biotechnology, Government of India: BT/PR40151/BTIS/137/5/2021.

### Competing Interests

The authors declare that they have no competing interests.

### Author Contributions

- Neetu Tyagi conceived and designed the experiments, performed the experiments, analyzed the data, prepared figures and/or tables, authored or reviewed drafts of the article, and approved the final draft.
- Shikha Roy conceived and designed the experiments, authored or reviewed drafts of the article, and approved the final draft.
- Dinesh Gupta conceived and designed the experiments, authored or reviewed drafts of the article, and approved the final draft.

### Data Availability

TCGA-GBM datasets are available at: https://portal.gdc.cancer.gov/projects/TCGA-GBM.

UCSC-Xena datasets are available at:
https://portal.gdc.cancer.gov/projects/TCGA-COAD.

lncRNACNVIntegrateR is available at GitHub, and Zenodo:
- https://github.com/tbgicgeb/lncRNACNVIntegrateR,
- tbgicgeb. (2025). tbgicgeb/lncRNACNVIntegrateR: v1.0.0 lncRNACNVIntegrateR (v1.0.0). Zenodo. https://doi.org/10.5281/zenodo.15861906.

Code is available in the Supplemental Files.

## Supplemental Information

Supplemental information for this article can be found online at http://dx.doi.org/10.7717/peerj.20131#supplemental-information.

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
