# Peer review of "LncRNACNVIntegrateR: a novel framework for correlating long non-coding RNAs with copy number variation abnormalities and disease progression"

_PeerJ, doi:10.7717/peerj.20131_

## Round 0.1 · original submission · Major Revisions

The manuscript describes a newly developed computational package that is potentially useful for the community. I ask the Authors to respond to the Reviewers' concerns so that the text can be accepted for publication.

·

Basic reporting

The study presents a useful R package for linking lncRNAs with CNV abnormalities and demonstrates prognostic value in GBM. While the methodology is technically sound, key improvements are needed to enhance reproducibility, clarity, and biological relevance.

Experimental design

1. Expand Validation with a Second Disease Case Study. Colon cancer (COAD): A well-studied cancer with known lncRNA-CNV interactions (e.g., CCAT1-LMYC, NEAT1). Demonstrates framework applicability beyond glioblastoma and leverages established lncRNA survival biomarkers.
2. Figure 1B and C: Redesign survival curves and plots with larger fonts, contrasting colors, and simplified annotations. Provide high-resolution versions in supplementary materials.

Validity of the findings

3. Explicitly state software versions in Methods: (e.g., PredictABEL R 1.7.0).

Additional comments

4. Divide Figure 1 into 3 figures using the blue outline
5. Discuss limitations in identifying causal vs. passenger CNV-lncRNA relationships

Reviewer 2 ·

Basic reporting

1. This manuscript tries to explain and describe the capabilities of LncRNACNVIntegrateR, a framework that was claimed to be novel
2. The structure of the manuscript is logical, and the language used is easily understood
3. The use case example used GBM data from the TCGA database
4. However, I think the manuscript is too brief and short, lacking details for the reproduction of the work. The details should be included.

Experimental design

1. It was not clearly mentioned how the GBM data was sorted and processed. Hence, the experimental design was not clear
2. The way the R analysis framework was developed was also not clearly stated
3. I suggest the authors write up clearly how the use case example was processed step by step so that readers can replicate and validate the work and results
4. Only one case study example was shown, so it raises the question of whether this framework is general enough for other datasets. A second case study example should be included.

Validity of the findings

1. Findings were not compared to other analysis/R packages, and only one use case example was shown; hence, the validity can be questioned
2. Please include the comparison of the findings using other methods/reports

Additional comments

1. The GitHub tutorial and documentation need to be improved as well, so that the package can be utilized by a wide audience
2. The GitHub link on the Docker Hub website is not correct and needs to be updated

Annotated reviews are not available for download in order to protect the identity of reviewers who chose to remain anonymous.

---

## Round 0.2 · accepted · Accept

Based on Reviewer's asessment and on my own reading, I'm pleased to accept the manuscript for further processing steps toward publication. Congratulations! I hope that the community will include your method in their data analysis pipelines.

Reviewer 2 ·

Basic reporting

No comment.

Experimental design

The authors have addressed the previously identified issues.

Validity of the findings

The two example use cases show the validity of the R scripts.

Additional comments

I'm satisfied with the corrections made by the authors.